# RANK-ADAPTIVE SPECTRAL PRUNING OF CONVOLUTIONAL LAYERS DURING TRAINING

## ABSTRACT

The computing cost and memory demand of deep learning pipelines have grown fast in recent years and thus a variety of techniques have been developed to reduce model parameters. The majority of these techniques focus on reducing inference costs by pruning the network after a pass of full training. A smaller number of methods addresses the reduction of training costs, mostly based on compressing the network via low-rank layer factorizations. Despite their efficiency for linear layers, these methods fail to effectively handle convolutional filters. In this work, we propose a low-parametric training method that factorizes the convolutions into tensor Tucker format and adaptively prunes the Tucker ranks of the convolutional kernel during training. Leveraging fundamental results from geometric integration theory of differential equations on tensor manifolds, we obtain a robust training algorithm that provably approximates the full baseline performance and guarantees loss descent. A variety of experiments against the full model and alternative low-rank baselines are implemented, showing that the proposed method drastically reduces the training costs, while achieving high performance, comparable to or better than the full baseline, outperforming competing low-rank approaches.

## 1 INTRODUCTION

A main limitation of state-of-the-art neural networks is their memory consumption and computational costs for inference and, especially, training. Leveraging well-known parameter redundancies (Cheng et al., 2015; Blalock et al., 2020; Frankle & Carbin, 2019), a large body of research work has been dedicated to removing redundant information from weights to reduce the memory and computational footprints. Such efforts include weight sparsification (Guo et al., 2016; Molchanov et al., 2017; He et al., 2017) and quantization (Wu et al., 2016; Courbariaux et al., 2016). Despite their considerably reduced resource requirements for inference, these methods struggle to achieve memory reduction during training. As pointed out in (Frankle & Carbin, 2019), training sparse neural networks from the start is challenging, and re-training accurate sparse architectures obtained through pruning with random initialization is commonly impossible. At the same time, as model and data size grow, the training phase of modern architectures can require several days on several hundreds of GPUs (Baker et al., 2022). Thus, being able to reduce the resource demand of the training phase while maintaining model performance is of critical importance.

A successful approach to reduce training parameters is based on low-rank factorizations. Instead of training full-weight matrices and pruning in a subsequent step, low-rank factorization methods use only low-rank parameter matrices during the entire training phase (Wang et al., 2021a; Khodak et al., 2021; Schotthöfer et al., 2022). Low-rank training of feed-forward fully-connected networks can reduce training costs by more than $90\%$, while maintaining approximately the same accuracy of the full model (Schotthöfer et al., 2022; Wang et al., 2021a; Khodak et al., 2021). However, these methods struggle to efficiently handle convolutional layers. In fact, efficiently compressing convolutional layers is a major challenge for most pruning and parameter reduction techniques (Cheng et al., 2018; Mishra et al., 2020). Another main challenge of low-rank training techniques is the dependence of the training convergence on the conditioning of the weight matrices (Schotthöfer et al., 2022) due to the high curvature of the low-rank manifold (Koch & Lubich, 2007), which may result in slow convergence rates and poor performance (Wang et al., 2021a; Khodak et al., 2021; Schotthöfer et al., 2022; Lebedev et al., 2015).

## 1.1 CONTRIBUTION

In this work, we introduce a novel low-rank training algorithm that directly aims at bridging the above gaps by efficiently handling convolutional layers. Available low-rank training methods (Wang et al., 2021a; Khodak et al., 2021; Schotthöfer et al., 2022; Idelbayev & Carreira-Perpinan, 2020; Yang et al., 2020) often treat convolutional layers by performing a matricization of the convolutional tensors. However, this procedure is computationally costly and fails to properly capture relevant low-rank components, resulting in poor compression performance. In this work, we propose a training algorithm that maintains the tensor structure unchanged and directly trains low-rank components in Tucker format. The proposed algorithm uses recent advances in low-rank approximation of differential equations (Ceruti et al., 2022; Ceruti & Lubich, 2022) to adjust the rank of the tensors while escaping the high-curvature of the low-rank Tucker manifold resulting in two key properties: **rank-adaptivity**, as the ranks of the convolutional layers change are automatically chosen during the epochs to match a desired compression rate, and **robustness**, as the convergence rate of the method does not deteriorate when the trained convolutional layers become ill-conditioned. To our knowledge, this is the first algorithm for training convolutional layers in tensor format which has these two fundamental properties. We provide a thorough rigorous analysis of the algorithm proving both the two properties above and showing that the computed low-rank Tucker network well-approximates the ideal full model. The theoretical findings are validated by an extensive experimental evaluation on different architectures and datasets, showing that the proposed method yields remarkable training compression rates (e.g. more than $95\%$ for VGG16 on CIFAR10), while achieving comparable or even better accuracy performance than the full baseline and alternative low-rank factorization strategies.

## 1.2 RELATED WORK

Related work on network compression methods differs structurally by the mathematical object of consideration, i.e. matrix- or tensor-valued parameter structures, as well as the type of parameter reduction. Weight pruning (Han et al., 2015; Narang et al., 2017; Ullrich et al., 2017; Molchanov et al., 2017; Wang et al., 2021b) enables parameter reduction by enforcing sparsity, i.e. zero-valued weights, whereas low-rank compression imposes parameter reduction by factorization of weight matrices (Idelbayev & Carreira-Perpinan, 2020; Li et al., 2019; Wang et al., 2021a) and tensors (Lebedev et al., 2015; Song et al., 2020; Astrid & Lee, 2017; Phan et al., 2020; Kossaifi et al., 2019; Kim et al., 2016; Kossaifi et al., 2019; Stoudenmire & Schwab, 2016). On top of approaches that transform tensor layers into compressed matrices (Schotthöfer et al., 2022; Idelbayev & Carreira-Perpinan, 2020; Li et al., 2019; Wang et al., 2021a), different tensor decompositions have been used to compress convolutional layers. Such approaches include CP decomposition (Lebedev et al., 2015; Song et al., 2020; Astrid & Lee, 2017; Phan et al., 2020), Tucker (Kossaifi et al., 2019; Kim et al., 2016), tensor trains (Kossaifi et al., 2019; Stoudenmire & Schwab, 2016) or a combination of these (Gabor & Zdunek, 2023). Other methods consider only the floating point representation of the weights, e.g. (Vanhoucke et al., 2011; Gong et al., 2015; Gupta et al., 2015; Courbariaux et al., 2015; Venkatesh et al., 2017), or a combination of the above (Liu et al., 2015). From the algorithmic point of view, related work can be categorized into methods that compress networks entirely in a postprocessing step after full-scale training (Nagel et al., 2019; Mariet & Sra, 2016; Lebedev et al., 2015; Kim et al., 2016; Gabor & Zdunek, 2023; Astrid & Lee, 2017), iterative methods where networks are pre-trained and subsequently compressed and fine-tuned (Han et al., 2015; Idelbayev & Carreira-Perpinan, 2020; Wang et al., 2021a), and methods that directly compress networks during training (Schotthöfer et al., 2022; Narang et al., 2017). As no full-scale training is needed, the latter approach offers the highest potential reduction of the overall computational footprint.

Only a few of these methods propose strategies for dynamically choosing the compression format during training or fine-tuning, e.g. by finding the ranks via alternating, constraint optimization in discrete (Li & Shi, 2018) and discrete-continuous fashions (Idelbayev & Carreira-Perpinan, 2020). However, both these approaches require knowledge of the full weights during training and overall are more computationally demanding than standard training. In (Schotthöfer et al., 2022), a rank-adaptive evolution of the gradient flow on a low-rank manifold was proposed to train and compress networks without the usage of the full-weight representation, however only for matrix-valued layers. The development of rank-adaptive training methods for tensor-valued layers poses non-trivial challenges that may prevent loss descent and performance of the compressed net. For example, numerical

instabilities arising from the CP decomposition during training have been observed in (Lebedev et al., 2015), and in (Phan et al., 2020).

## 2 LOW-RANK TUCKER REPRESENTATION OF CONVOLUTIONAL LAYERS

Neural networks' convolutional filters are the backbone of many groundbreaking machine learning architectures. These layers are defined by a convolutional kernel which consists of a four-mode tensor $W \in \mathbb{R}^{n_1 \times n_2 \times n_3 \times n_4}$, where $n_1$ is the number of output channels, $n_2$ is the number of input channels and $(n_3, n_4)$ are the spatial dimensions of the filter. A kernel represents $n_1$ convolutional filters of shape $n_2 \times n_3 \times n_4$, which are applied to the input embedding tensor $Z \in \mathbb{R}^{N \times n_2 \times N_1 \times N_2}$, where $N$ is the batch size and $(N_1, N_2)$ are the spatial dimensions of the embedding's channel. The convolution operation $W * Z$ is then defined as

$$(W * Z)(i_1, i_2, i_3, i_4) = \sum_{j_2=1}^{n_2} \sum_{j_3=1}^{n_3} \sum_{j_4=1}^{n_4} W(i_2, j_2, j_3, j_4) Z(i_1, j_2, i_3 - j_3, i_4 - j_4). \quad (1)$$

Convolutions are linear operations on tensor blocks. To enable the use of matrix techniques, a standard approach to handle convolutional layers is via a matricization step, in which the tensor $W$ is reshaped into a matrix $\widetilde{W} \in \mathbb{R}^{n_1 \times n_2 n_3 n_4}$ and the input embedding $Z$ into a third-order tensor $\widetilde{Z} \in \mathbb{R}^{N \times n_2 n_3 n_4 \times L}$ obtained by stacking $L$ blocks of the vectorized version of $Z$ following the sliding patterns of the kernel. Such a matrix representation allows, in particular, to compress the convolution using a low-rank matrix factorization, an approach that is widely adopted by the recent literature (Schotthöfer et al., 2022; Idelbayev & Carreira-Perpinan, 2020; Yang et al., 2020; Wang et al., 2021a; Khodak et al., 2021). However, this matricization operation is too restrictive as a representation as it destroys the structure of the convolutional kernel and does not allow capturing compressible modes in the higher-order sense. For this reason, most available low-rank pruning techniques have struggled to reduce the memory footprint of convolutional layers by using a low-rank approximation of $\widetilde{W}$, see e.g. (Schotthöfer et al., 2022; Li et al., 2019; Khodak et al., 2021). On the other hand, it is well-known that low-rank decompositions based on higher-order tensor ranks can provide much better representations of compressed convolutional layers (Stoudenmire & Schwab, 2016; Song et al., 2020; Gabor & Zdunek, 2023; Li et al., 2019; Kossaifi et al., 2019).

In this paper, we use the Tucker decomposition to represent low-rank convolutional layers in tensor format. This representation maintains the convolutional structure and represents the rank of each mode in the filter individually. Moreover, the Tucker format forms a smooth Riemannian manifold. Exploiting this geometric property we design an algorithm that trains using only the low-rank factors, while simultaneously adjusting the Tucker ranks. Below we review the Tucker representation of a convolutional layer and we present the new training algorithm next.

### 2.1 TUCKER REPRESENTATION OF A CONVOLUTIONAL FILTER

For a tensor $W$ we write $\mathrm{Mat}_i(W)$ to denote the matrix obtained by unfolding $W$ along its $i$-th mode. The tuple $\rho = (r_1, r_2, \ldots, r_d)$ is called Tucker rank of $W$ if $r_i = \mathrm{rank}(\mathrm{Mat}_i(W))$. Every fourth-order tensor $W$ with Tucker rank $\rho = (r_1, \ldots, r_4)$ can be written in Tucker form (or Tucker decomposition) $W = C \times_1 U_1 \times_2 U_2 \times_3 U_3 \times_4 U_4 = C \times_{i=1}^4 U_i$, entrywise defined as

$$W(i_1, i_2, i_3, i_4) = \sum_{\alpha_1=1}^{r_1} \cdots \sum_{\alpha_4=1}^{r_4} C(\alpha_1, \alpha_2, \alpha_3, \alpha_4) U_1(i_1, \alpha_1) U_2(i_2, \alpha_2) U_3(i_3, \alpha_3) U_4(i_4, \alpha_4), \quad (2)$$

where $C \in \mathbb{R}^{r_1 \times \cdots \times r_4}$ is a *core tensor* of full Tucker rank $\rho = (r_1, \ldots, r_4)$ and the $U_i \in \mathbb{R}^{n_i \times r_i}$ are matrices with orthonormal columns. Note that in terms of the $i$-th unfolding, equation 2 reads $\mathrm{Mat}_i(W) = U_i S_i V_i^\top$ with $S_i = \mathrm{Mat}_i(C)$ and $V_i = \otimes_{j \neq i} U_j$, i.e. the usual matrix decomposition.

Using this decomposition, the convolution with kernel $W$ can be written completely in terms of the factors $U_i$ and the core $C$. In fact, if we let

$$A(\alpha_2, \alpha_3, \alpha_4, i_1, i_3, i_4) = \sum_{j_2, j_3, j_4} U_2(j_2, \alpha_2) U_3(j_3, \alpha_3) U_4(j_4, \alpha_4) Z(i_1, j_2, i_3 - j_3, i_4 - j_4)$$

where $A = (U_2 \otimes U_3 \otimes U_4) * Z$ is the convolution with the factorized kernel $(U_2 \otimes U_3 \otimes U_4)$, then

$$(W * Z)(i_1, i_2, i_3, i_4) = \sum_{\alpha_1, \ldots, \alpha_4} C(\alpha_1, \alpha_2, \alpha_3, \alpha_4) A(\alpha_2, \alpha_3, \alpha_4, i_1, i_3, i_4) U_1(i_2, \alpha_1),$$

which we can compactly write as

$$W * Z = C \times_{2,3,4} [(U_2 \otimes U_3 \otimes U_4) * Z] \times_1 U_1.$$

From this representation, we immediately see that if $W$ is represented in Tucker format, then the cost of storing $W$ and of performing the convolution operation $W * Z$ is $O(r_1 r_2 r_3 r_4 + n_1 r_1 + n_2 r_2 + n_3 r_3 + n_4 r_4)$, as opposed to the $O(n_1 n_2 n_3 n_4)$ cost required by the standard full representation. Clearly, when $n_i \gg r_i$, for example $n_i > 1.5 r_i$, the latter is much larger than the former. As an example, for Alexnet trained on Cifar10 the third convolutional layer has Tucker ranks $[r_1, \ldots, r_4] = [76, 76, 3, 3]$ (with $[n_1, \ldots, n_4] = [256, 256, 3, 3]$). Thus, we have that the memory complexity of the compressed Tucker format is $5.2 \times 10^4$ versus $5.89 \times 10^5$ elements to be stored, i.e., compression of approximately $91.3\%$ on that layer (Appendix A in the supplementary material and Section 4.2).

## 3 DYNAMICAL LOW-RANK TRAINING OF CONVOLUTIONS IN TUCKER FORMAT

For $\rho = (r_1, r_2, r_3, r_4)$, the set $\mathcal{M}_\rho = \{W : \mathrm{rank}(\mathrm{Mat}_i(W)) = r_i, \ i = 1, \ldots, 4\}$ is a manifold with the following tangent space at any point $W = C \times_{i=1}^4 U_i \in \mathcal{M}_\rho$ (Koch & Lubich, 2010)

$$T_W \mathcal{M}_\rho = \left\{ \delta C \mathop{\times}_{i=1}^4 U_i + \sum_{j=1}^4 C \times_j \delta U_j \mathop{\times}_{k \neq j} U_k : \delta C \in \mathbb{R}^{r_1 \times \cdots \times r_4}, \ \delta U_j \in T_{U_j} \mathcal{S}_j \right\} \tag{3}$$

where $\mathcal{S}_j$ is the Stiefel manifold of real $n_i \times r_i$ matrices with orthonormal columns. To design a strategy that computes convolutional filters within $\mathcal{M}_\rho$ using only the low-rank Tucker factors $C$ and $\{U_i\}_i$, we formulate the training problem as a continuous-time gradient flow projected onto the tangent space equation 3. As shown in Section 3.2, the continuous formulation will allow us to derive a modified backpropagation pass which uses only the individual small factors $C, \{U_i\}_i$ and that does not suffer from a slow convergence rate due to potential ill-conditioned tensor modes (see also Section 4.3). Moreover, it will allow us to prove a global approximation result showing convergence towards a "low-Tucker-rank winning ticket": a subnetwork formed of convolutional layers with a low Tucker rank that well-approximates the original full model.

Let $f$ be a convolutional neural network and let $W$ be a convolutional kernel tensor within $f$. Consider the problem of minimizing the loss function $\mathcal{L}$ with respect to just $W$, while maintaining the other parameters fixed. This problem can be equivalently formulated as the differential equation

$$\dot{W}(t) = -\nabla_W \mathcal{L}(W(t)) \tag{4}$$

where, for simplicity, we write the loss as a function of only $W$ and where "dot" denotes the time derivative. When $t \to \infty$, the solution of equation 4 approaches the desired minimizer. Now, suppose we parametrize each convolutional layer in a time-dependent Tucker form $W(t) = C(t) \times_{i=1}^4 U_i(t)$. Using standard derivations from dynamical model order reduction literature (Koch & Lubich, 2010), we derive below the equations for the individual factors $C(t)$ and $U_i(t)$.

To this end, notice that by definition, if $W(t) \in \mathcal{M}_\rho$, then $\dot{W}(t) \in T_{W(t)} \mathcal{M}_\rho$, the tangent space of $\mathcal{M}_\rho$ at $W(t)$. Thus, when $W(t) \in \mathcal{M}_\rho$, equation 4 boils down to (Koch & Lubich, 2010)

$$\dot{W}(t) = -P(W(t)) \nabla_W \mathcal{L}(W(t)) \tag{5}$$

where $P(W)$ denotes the orthogonal projection onto $T_W \mathcal{M}_\rho$. Note that, for a fixed point $W \in \mathcal{M}_\rho$, $P(W)$ is a linear map defined in terms of the optimization problem

$$P(W)\left[-\nabla_W \mathcal{L}(W)\right] = \operatorname*{argmin}_{\delta W \in T_W \mathcal{M}_\rho} \|\delta W + \nabla_W \mathcal{L}(W)\|, \tag{6}$$

where here and henceforth $\|\cdot\|$ denotes the Frobenius norm. Since $\|\cdot\|$ is induced by the inner product $\langle W, Y \rangle = \sum_{i_1, \ldots, i_4} W(i_1, \ldots, i_4) Y(i_1, \ldots, i_4)$, the projection in equation 6 can be derived by imposing orthogonality with any element of the tangent space. This means that it is possible to

derive the ODEs for the projected dynamics by imposing a time-dependent Galerkin condition on the tangent space of $\mathcal{M}_\rho$, i.e. by imposing the following

$$\langle \dot{W}(t) + \nabla_W \mathcal{L}(W(t)), \delta W \rangle = 0, \qquad \forall \, \delta W \in T_{W(t)} \mathcal{M}_\rho \,.$$

Using the representation in equation 3 combined with the standard gauge conditions $U_i^\top \delta U_i = 0$, $\forall \delta U_i \in T_{U_i} \mathcal{S}_i$, for the Stiefel manifold $\mathcal{S}_i$, we obtain that the projected gradient flow in equation 5 coincides with the coupled matrix-tensor system of ODEs

$$\begin{cases} \dot{U}_i &= -(I - U_i U_i^\top) \operatorname{Mat}_i\big(\nabla_W \mathcal{L}(W) \times_{j \neq i} U_j^\top\big) \operatorname{Mat}_i(C)^\dagger, \quad i = 1, \dots, 4 \\ \dot{C} &= -\nabla_W \mathcal{L}(W) \times_{j=1}^4 U_j^\top \,. \end{cases} \tag{7}$$

where $\dagger$ denotes the pseudoinverse and where we omitted the dependence on $t$ for brevity. Even though equation 7 describes the dynamics of the individual factors, the equations for each factor are not fully decoupled. In fact, a direct integration of equation 7 would still require taping the gradients $\nabla_W \mathcal{L}$ with respect to the full convolutional kernel $W$. Moreover, the presence of the pseudoinverse of the matrices $\operatorname{Mat}_i(C)^\dagger$ adds a stiffness term to the differential equation, making its numerical integration unstable. The presence of this stiff term is actually due to the intrinsic high-curvature of the manifold $\mathcal{M}_\rho$ and is well understood in the dynamic model order reduction community (Koch & Lubich, 2007; Lubich & Oseledets, 2014; Kieri et al., 2016; Lubich et al., 2018; Ceruti & Lubich, 2022; Ceruti et al., 2022). As observed in (Schotthöfer et al., 2022), an analogous term arises when looking at low-rank matrix parameterizations, and it is responsible for the issue of slow convergence of low-rank matrix training methods which is observed in (Wang et al., 2021a; Khodak et al., 2021; Schotthöfer et al., 2022).

To overcome these issues, we make the following key change of variable. Let $\operatorname{Mat}_i(C)^\top = Q_i S_i^\top$ be the QR decomposition of $\operatorname{Mat}_i(C)^\top$. Note that $S_i$ is a small square invertible matrix of size $r_i \times r_i$. Then, the matrix $K_i = U_i S_i$ has the same size as $U_i$ and spans the same vector space. However, the following key result holds for $K_i$.

**Theorem 1.** *Let $W = C \times_{i=1}^4 U_i \in \mathcal{M}_\rho$ be such that equation 6 holds. Let $\operatorname{Mat}_i(C)^\top = Q_i S_i^\top$ be the QR decomposition of $\operatorname{Mat}_i(C)^\top$ and let $K_i = U_i S_i$. Then,*

$$\dot{K}_i = -\nabla_{K_i} \mathcal{L}\big(\operatorname{Ten}_i(Q_i^\top) \times_{j \neq i} U_j \times_i K_i\big) \qquad and \qquad \dot{C} = -\nabla_C \mathcal{L}(C \times_{j=1}^4 U_j) \tag{8}$$

*where $\operatorname{Ten}_i$ denotes "tensorization along mode $i$", i.e. the inverse reshaping operation of $\operatorname{Mat}_i$.*

The proof is provided in Appendix B in the supplementary material. The theorem above allows us to simplify equation 7 obtaining a gradient flow that only depends on the small matrices $K_i$ and the small core tensor $C$. Moreover, it allows us to get rid of the stiffness term, as no inversion is now involved in the differential equations. We would like to underline the importance of the careful construction of $K_i$ to arrive at this Theorem, as unlike a naive extension of (Schotthöfer et al., 2022) to Tucker tensors, our construction does not require computational costs of $O(n_i \Pi_{j \neq i} r_j^2)$ which would render the resulting training method impractical.

Based on Theorem 1, we formulate in the next section the proposed modified training step for convolutional layers. Notably, by a key basis-augmentation step, we will equip the algorithm with a rank-adjustment step that learns the Tucker rank of the convolutions during training, while maintaining guarantees of descent and approximation of the full convolutional kernel, as shown by the theoretical analysis in Section 3.2.

## 3.1 RANK-ADAPTIVE ALGORITHM AND IMPLEMENTATION DETAILS

The training algorithm for convolutional layers in Tucker format is presented in Algorithm 1. Each time we back-propagate through a convolutional layer $W = C \times_{i=1}^4 U_i$, we form the new variable $K_i = U_i S_i$ as in Theorem 1, we integrate the ODE in equation 9 from $K_i(0) = K_i$ to $K_i(\lambda)$, $\lambda > 0$, and then update the factors $U_i$ by forming an orthonormal basis of the range of $K_i(\lambda)$. In practice, we implement the orthonormalization step via the QR factorization, while we perform the integration of the gradient flow via stochastic gradient descent with momentum and learning rate $\lambda$, which coincides with a stable two-step linear multistep integration method (Scieur et al., 2017). Once all the factors $U_i$ are updated, we back-propagate the core term by integrating the equation for $C$ in equation 9, using the same approach.

---

**Algorithm 1:** TDLRT: Dynamical Low-Rank Training of convolutions in Tucker format.

---

**Input :** Convolutional filter $W \sim n_1 \times n_2 \times n_3 \times n_4$;
  Initial low-rank factors $C \sim r_1 \times \cdots \times r_4$; $U_i \sim n_i \times r_i$;
  `adaptive`: Boolean flag that decides whether or not to dynamically update the ranks;
  $\tau$: singular value threshold for the adaptive procedure.

**1 for** *each mode $i$* **do**
**2** $\quad$ $Q_i S_i^\top \leftarrow$ QR decomposition of $\mathrm{Mat}_i(C)^\top$
**3** $\quad$ $K_i \leftarrow U_i S_i$
**4** $\quad$ $K_i \leftarrow$ descent step; direction $\nabla_{K_i} \mathcal{L}(\mathrm{Ten}_i(Q_i^\top) \times_{j \neq i} U_j \times_i K_i)$; starting point $K_i$
**5** $\quad$ **if** `adaptive` **then** $\hspace{4cm}$ /* Basis augmentation step */
**6** $\quad\quad$ $\mid$ $K_i \leftarrow [K_i \,|\, U_i]$
**7** $\quad$ $U_i^{\mathrm{new}} \leftarrow$ orthonormal basis for the range of $K_i$
**8** $\widetilde{C} \leftarrow C \times_{i=1}^{4} (U_i^{\mathrm{new}})^\top U_i$
**9** $C \leftarrow$ descent step; direction $\nabla_C \mathcal{L}\big(\widetilde{C} \times_{i=1}^{4} U_i^{\mathrm{new}}\big)$; starting point $\widetilde{C}$

**10 if** `adaptive` **then** $\hspace{4cm}$ /* Rank adjustment step */
**11** $\quad$ $(C, U_1, \ldots, U_4) \leftarrow$ Tucker decomposition of $C$ up to relative error $\tau$
**12** $\quad$ $U_i \leftarrow U_i^{\mathrm{new}} U_i$, for $i = 1, \ldots, 4$
**13 else**
**14** $\quad$ $U_i \leftarrow U_i^{\mathrm{new}}$, for $i = 1, \ldots, 4$

---

An important feature of the proposed back-propagation step is that the Tucker rank of the new kernel can be adaptively learned with a key basis-augmentation trick: each time we backprop $K_i \in \mathbb{R}^{n_i \times r_i}$, we form an augmented basis $\widetilde{K}_i$ by appending the previous $U_i$ to the new $K_i(\lambda)$, $\widetilde{K}_i = [K_i | U_i]$. We compute an orthonomal basis $U_i^{\mathrm{new}} \in \mathbb{R}^{n_i \times 2r_i}$ for $\widetilde{K}_i$ and we form the augmented $2r_1 \times \cdots \times 2r_4$ core $\widetilde{C} = C \times_{i=1}^{4} (U_i^{\mathrm{new}})^\top U_i$. We then backpropagate the core $C$ integrating equation 9 starting from $C(0) = \widetilde{C}$. Finally, we perform a rank adjustment step by computing the best Tucker approximation of $\widetilde{C}$ to a relative tolerance $\tau > 0$. This step corresponds to solving the following optimization (rounding) task:

$$\text{Find } \widehat{C} \in \mathcal{M}_{\leq 2\rho} \text{ of smallest rank } \rho' = (r_1', \ldots, r_4') \text{ such that } \quad \|\widetilde{C} - \widehat{C}\| \leq \tau \|\widetilde{C}\|$$

where $\rho = (r_1, \ldots, r_4)$ and $\mathcal{M}_{\leq 2\rho}$ denotes the set of tensors with component-wise Tucker rank lower than $2\rho$. In practice, this is done by unfolding the tensor along each mode and computing a truncated SVD of the resulting matrix, as implemented in the `tntorch` library (Usvyatsov et al., 2022). The tensor $\widehat{C} \in \mathcal{M}_{\rho'}$ is then further decomposed in its Tucker decomposition yielding a factorization $\widehat{C} = C' \times_{i=1}^{4} U_i' \in \mathcal{M}_{\rho'}$. The parameter $\tau$ is responsible for the compression rate of the method, as larger values of $\tau$ yield smaller Tucker ranks and thus higher parameter reduction. To conclude, the computed $U_i' \in \mathbb{R}^{2r_i \times r_i'}$ with $r_i' \leq 2r_i$ are then pulled back to the initial dimension of the filter by setting $U_i = U_i^{\mathrm{new}} U_i' \in \mathbb{R}^{n_i \times r_i'}$, and the new core tensor $C$ is then assigned $C'$. performance

### 3.2 ANALYSIS OF LOSS DESCENT AND SEARCH OF WINNING TICKETS

In this section, we present our main theoretical results. First, we show that the back-propagation step in Algorithm 1 guarantees descent of the training loss, provided the compression tolerance is not too large. Second, we show that the convolutional filter in compressed Tucker format computed via the rank-adaptive back-propagation step in Algorithm 1 well-approximates the full filter that one would obtain by standard training, provided the gradient flow of the loss is, at each step, approximately low-rank. For smooth losses $\mathcal{L}$, the latter requirement can be interpreted as assuming the existence of a winning ticket of low Tucker rank, i.e. a subnetwork whose convolutions have small Tucker rank and that well-approximates the performance of the large initial net. In this sense, our second result shows that, if a low-Tucker-rank winning ticket exists, then the proposed Tucker-tensor flow equation 9 approaches it. This result provides the Tucker-tensor analogue of the approximation theorem presented in (Schotthöfer et al., 2022) for layers in matrix format and, while the two results are alike, we emphasize that the proof techniques differ significantly. One of the core differences

results from the construction of the $K_i$ terms which, unlike a straightforward extension of (Schotthöfer et al., 2022) to Tucker format, are designed via a carefully chosen factorization of the core tensor to extract the matrices $Q_i$ from $K_i$ and to avoid a prohibitive computational cost of $O(n_i \prod_{j \neq i} r_j^2)$. For the sake of brevity, some statements here are formulated informally and all proofs and details are deferred to Appendix C in the supplementary material.

Suppose that for each convolution $W$, the gradient $\nabla_W \mathcal{L}$, as a function of $W$, is locally bounded and Lipschitz, i.e. $\|\nabla_W \mathcal{L}(Y)\| \leq L_1$ and $\|\nabla_W \mathcal{L}(Y_1) - \nabla_W \mathcal{L}(Y_2)\| \leq L_2 \|Y_1 - Y_2\|$ around $W$. Then,

**Theorem 2.** *Let $W(\lambda) = C \times_{j=1}^4 U_j$ be the Tucker low-rank tensor obtained after one training iteration using Algorithm 1 and let $W(0)$ be the previous point. Assuming the one-step integration from 0 to $\lambda$ is done exactly, it holds $\mathcal{L}_W(W(\lambda)) \leq \mathcal{L}_W(W(0)) - \alpha\lambda + \beta\tau$, where $\alpha, \beta > 0$ are constants independent of $\lambda$ and $\tau$, and where $\mathcal{L}_W$ denotes the loss as a function of only $W$.*

**Theorem 3.** *For an integer $k$, let $t = k\lambda$, and let $W(t)$ be the full convolutional kernel, solution of equation 4 at time $t$. Let $C(t), \{U_i(t)\}_i$ be the Tucker core and factors computed after $k$ training steps with Algorithm 1, where the one-step integration from 0 to $\lambda$ is done exactly. Finally, assume that for any $Y$ in a neighborhood of $W(t)$, the gradient flow $-\nabla \mathcal{L}_W(Y)$ is "$\varepsilon$-close" to $T_Y \mathcal{M}_\rho$. Then,*

$$\|W(t) - C(t) \times_{j=1}^4 U_j(t)\| \leq c_1 \varepsilon + c_2 \lambda + c_3 \tau / \lambda$$

*where the constants $c_1$, $c_2$ and $c_3$ depend only on $L_1$ and $L_2$.*

In particular, both bounds in the above theorems do not depend on the higher-order singular values of the exact nor the approximate solution, which shows that the method does not suffer instability and slow convergence rate due to potential ill-conditioning (small higher-order singular values). Note that this result is crucial for efficient training on the low-rank manifold and is not shared by direct gradient descent training approaches as we will numerically demonstrate in the following section.

## 4 EXPERIMENTS

In the following, we conduct a series of experiments to evaluate the performance of the proposed method as compared to both full and low-rank baselines. The full baseline is the network trained via standard implementation. We then consider two low-rank baselines in tensor format: Canonic-Polyadic factorization, as done in e.g. (Lebedev et al., 2015; Song et al., 2020; Astrid & Lee, 2017; Phan et al., 2020); Tucker factorization, as in e.g. (Kossaifi et al., 2019; Kim et al., 2016). Further, we compare with low-rank training in matrix format, implemented after reshaping the convolutions, as done in e.g. (Idelbayev & Carreira-Perpinan, 2020; Li et al., 2019; Wang et al., 2021a; Khodak et al., 2021). All the methods above train each of the low-rank factors in the decomposition by implementing one pass of forward and backward propagation on each of the factors individually (and simultaneously) in a block-coordinate fashion. Additionally, we compare with the matrix DLRT algorithm, where the standard forward and backward passes are endowed with a rank-adaptive QR projection step, similar to the proposed Algorithm 1. In terms of pruning techniques based on sparsification, we compare with methods from two of the most popular strategies: iterative magnitude pruning (IMP) (Frankle & Carbin, 2019), and single-shot pruning at initialization, single-shot network pruning (SNIP) (Lee et al., 2019) and Gradient Signal Preservation (GraSP) (Wang et al., 2020). All the experiments are conducted using PyTorch and a single Nvidia RTX 3090 GPU. The code is available in the supplementary material.

### 4.1 COMPRESSION PERFORMANCE

The compression performance of TDLRT is evaluated on CIFAR10. For this dataset, the typical data augmentation procedure is employed: a composition of standardization, random cropping and a random horizontal flip of images is performed on training images. All methods are trained using a batch size of 128 for 70 epochs each, as done in (Wang et al., 2021a; Khodak et al., 2021). All the baseline methods are trained with the SGD optimizer; the starting learning rate of 0.05 is reduced by a factor of 10 on plateaus and momentum is chosen as 0.1 for all layers. The rank $\hat{r}$ of each tensor mode for the fixed-rank baseline methods is determined by a parameter $\kappa$, i.e. we set $\hat{r} = \kappa r_{\max}$. The proposed TDLRT method employs Algorithm 1, where SGD is used for the descent steps at lines 4 and 9, with momentum and learning rate as above. Dynamic compression during training is governed by the singular value threshold $\tau$, see Section 3.1.

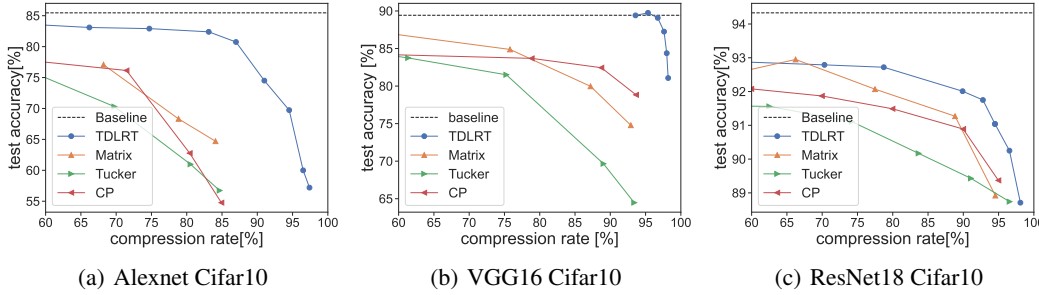

|                | (a) Alexnet Cifar10 | (b) VGG16 Cifar10 | (c) ResNet18 Cifar10 |
|---|---|---|---|

Figure 1: Comparison of compression performance for different models against the full baseline. Mean accuracy of 20 weight initializations is displayed. TDLRT achieves higher compression rates at higher accuracy with lower variance between initializations.

Table 1: Comparison of the best-performing compression rates for different methods on the CIFAR10 benchmark with Alexnet, VGG16, and ResNet18. TDLRT outperforms the factorization-based as well as the pruning-based baselines in terms of accuracy and compression rate.

|          |                   | VGG16 | | Alexnet | | ResNet18 | |
|----------|-------------------|---------------|----------|---------------|----------|---------------|----------|
|          |                   | test acc. [%] | c.r. [%] | test acc. [%] | c.r. [%] | test acc. [%] | c.r. [%] |
|          | Baseline          | 89.43         | 0.0      | 85.46         | 0.0      | 94.33         | 0.0      |
| Low-rank | TDLRT             | **89.59**     | **95.34** | **82.39**    | **83.12** | **92.72**    | 78.73    |
|          | Matrix DLRT       | 89.13         | 83.22    | 73.57         | 71.57    | 80.98         | 56.85    |
|          | Tucker-factorized | 83.74         | 61.34    | 70.3          | 69.74    | 91.11         | 74.19    |
|          | Matrix-factorized | 79.96         | 87.20    | 77.07         | 68.20    | 92.07         | 77.49    |
|          | CP-factorized     | 83.68         | 78.85    | 76.14         | 71.46    | 91.87         | 69.95    |
| Pruning  | SNIP              | 89.58         | 56.23    | –             | –        | 89.5          | 78.5     |
|          | IMP               | 87.21         | 58.54    | –             | –        | 90.5          | **82.50** |
|          | GraSP             | 88.5          | 77.3     | –             | –        | 89.4          | 77.9     |

Figure 1 shows the mean accuracy of TDLRT as compared to competing factorization baselines. TDLRT achieves higher compression rates at higher accuracy with lower variance between weight initializations than the competing methods. In the case of the VGG16 benchmark, TDLRT is able to maintain baseline accuracy for compression rates over $90\%$ and exceeds the baseline on average for $\tau = 0.03$, i.e. $95.3\%$ compression. Alexnet has $16.8\%$ of the parameters of VGG16, thus compression is naturally harder to achieve. Yet, TDLRT outperforms the baseline methods and remains close to the full network performance. Similar behaviour is observed also on ResNet18.

Table 1 shows a comparison of the best-performing compression between all the factorization-based and pruning-based baseline methods as well as TDLRT in the CIFAR10 benchmark for Alexnet, ResNet18, and VGG16. In the proposed evaluation, TDLRT is on par or outperforms all the alternatives, including pruning based on sparsity (implemented without warmup for the sake of a fair comparison) as well as the matrix-valued DLRT, due to the higher flexibility of the Tucker format, where compression along each tensor mode individually is possible. The compression rate (c.r.) is computed as $1 - c/f$, where $c$ is the number of parameters in the compressed model after training and $f$ is the number of parameters of the full model. While this is the compression rate after training, we emphasize that methods based on factorizations yield an analogous compression rate during the entire training process. We also remark that no DLRT version of CP decomposition is shown as CP is not suited for dynamical low-rank training due to its lack of a manifold structure.

## 4.2 COMPUTATIONAL PERFORMANCE

The computational performance in inference and training of convolutional layers in Tucker decomposition is dependent on their current tensor ranks, see Section 2.1. We evaluate the inference time of $120K$ RGB images and memory footprint of VGG and AlexNet in Tucker factorization as used in

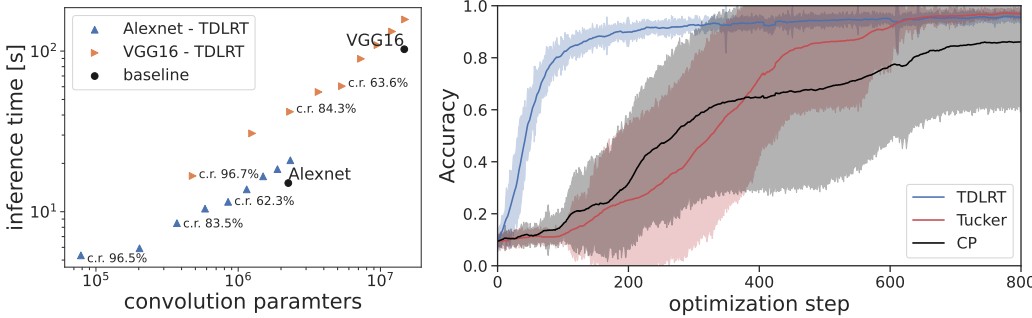

Figure 2: Left panel: Computational footprint of low-rank convolutions. TDLRT surpasses the baseline performance for meaningful compression rates. Right panel: Convergence behavior of Lenet5 on MNIST dataset in the case of an initial overestimation of the rank, with exponentially decaying singular values. Mean and standard deviation (shaded area) over 10 random initializations.

Algorithm 1 and compare them to the non-factorized baseline models in Figure 2. As a result, for realistic compression rates, see also Figure 1, the computational footprint of TDLRT is significantly lower than the corresponding baseline model.

## 4.3 ROBUSTNESS OF THE OPTIMIZATION

To further highlight the advantages of Algorithm 1 as compared to standard simultaneous gradient descent on the factors of the decomposition, we show in Figure 2 the accuracy history of LeNet5 on MNIST using TDLRT as compared to standard training on Tucker and CP decompositions. In the case of TDLRT, an optimization step denotes the evaluation of Algorithm 1 for all convolutional layers for one batch of training data, while for the other methods, we refer to a standard SGD batch update for all factors of the tensor decompositions of all layers. All linear layers of the network are trained with a traditional gradient descent update and are not compressed. In this experiment, we initialize the network weights to simulate a scenario where the rank is overestimated. To this end, we employ spectral initialization with singular values decaying exponentially with powers of ten. Integrating the low-rank gradient flow with the TDLRT Algorithm 1, leads to faster and more robust convergence rates of the network training process.

## 5 DISCUSSION AND LIMITATIONS

This work introduces an algorithm that adaptively reduces the Tucker rank of convolutional layers during training, reducing computational and memory costs. The method is supported by rigorous proofs of approximation to the full baseline and loss descent. Several tests validate the provided theoretical results and show faster and more robust convergence compared to baseline approaches, alongside better performance. Further, TDLRT adaptively learns the Tucker ranks without the need for matricization, increasing the approximation expressivity of the low-rank network.

Alongside several advantages and proven properties, TDLRT and tensor factorization methods for convolutional layers in general face several challenges. First, we notice that while providing a reduction in cost and memory corresponding to model parameters and optimizer, TDLRT does not aim at reducing activation costs, in the terminology of (Sohoni et al., 2019). Second, we notice that an efficient compression requires an appropriate strategy to choose the tolerance parameter $\tau$ combined with a computationally efficient tensor rounding algorithm at line 11 of Algorithm 1. Even if `tntorch` provides an efficient approach, the application of alternative techniques based on, e.g., randomization, may lead to a boost in computational performance. Finally, we remark that TDLRT relies on the assumption that well-performing low-rank Tucker sub-nets exist in the reference network. While we observe this empirically, further investigations are required to provide theoretical evidence in support of this assumption, similar to what is done in the case of fully-connected linear layers in e.g. deep linear networks (Arora et al., 2019; Bah et al., 2022).

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
