# OpenReview forum: "Rank-adaptive spectral pruning of convolutional layers during training"
_ICLR.cc/2024/Conference — Submitted to ICLR 2024_

### Official Review · Reviewer_SS8V · 2023-11-10

**Soundness:** 2 fair
**Presentation:** 3 good
**Contribution:** 2 fair
**Rating:** 3
**Confidence:** 4

**Summary:**

This paper proposes a dynamic low-rank training method for convolutional neural networks that factorizes convolutions into tensor Tucker format and adaptively prunes the Tucker ranks of the convolutional kernel during training. The proposed method drastically reduces training costs while achieving high performance, outperforming competing low-rank approaches. The paper also includes a discussion of the geometric integration theory of differential equations on tensor manifolds used in this work. Empirical evaluation on CIFAR10 using VGG and Alexnet models also shows the effectiveness of the proposed method.

**Strengths:**

1. The paper is well-written and easy to follow. The authors provide a clear motivation and a nice introduction to the problem.

2. The connection to recent advance in dynamic low-rank approximation is interesting and indeed a nice direction to apply it to dynamic low-rank training.

3. Strong theoretical understanding on the proposed method. The authors provide a clear derivation of TDLRT and its convergence analysis.

**Weaknesses:**

1. Concerns on novelty. The proposed TDLRT is clearly an extension of matrix DLRT [1] on Tensor format. Although the authors emphasize this extension is non-trivial as it requires the use of Tucker decomposition and additional techniques for theoretical analysis, I believe the extension is not significant enough to be considered as a novel contribution.

2. The paper lacks discussion on computational complexity during training. The authors claim the proposed method drastically reduces the training cost but the empirical results on training cost are missing. In Figure 2, the authors only show the computational footprint of the inference stage. However, for such dynamical compression method I believe training cost is more important to discuss. It would be great to show a figure similar to Figure 2 in [1] to visualize the training cost across different rank values. In Table 1, it also would be great to present the total training cost by different methods for a fair comparison.

3. More details of proposed method are needed. For example, what initialization on low-rank factors and does it affect the performance? How to choose the threshold?

4. Evaluation on large-scale datasets and SOTA models are needed. CIFAR-10 is too small to show the effectiveness of the proposed method, and VGG16 and Alexnet are outdated models. It would be great to evaluate methods on ImageNet at least using ResNet50.

[1] Steffen Schotthöfer, Emanuele Zangrando, Jonas Kusch, Gianluca Ceruti, and Francesco
Tudisco. Low-rank lottery tickets: finding efficient low-rank neural networks via ma-
trix differential equations. In Advances in Neural Information Processing Systems,
2022.

**Questions:**

1. Would be great to provide more details on the training behavior of TDLRT, such as how rank evolves during training, convergence behaviors regarding various compression thresholds, etc.

2. How often do you perform basis augmentation and tucker decomposition using SVD? If it is performed frequently for every step, does the numerical instability affects the convergence? such that, at the later stage of training, the perturbation on $C$ by learning algorithms may be too small compared to the numerical error induced by SVD.

3. Is there any way to further improve efficiency to make the method practical?

---

> ### Author Response · Authors · 2023-11-15
>
> Dear Reviewer, thank you for your feedback.
>
> 1. You state in your review that you "believe the extension is not significant enough to be considered as a novel contribution". We are unsure why you came to this conclusion and kindly ask you to please detail the reasons for this evaluation.
> We emphasize that the proposed method is conceptually different from that in [1]; it requires nontrivial additions and ideas for the mathematical analysis *and* the algorithm itself; and it performs significantly better than DLRT in [1]. Please notice that we have provided a detailed explanation on novelty in the general comment above.
> As we pointed out in our paper, the proof of our Thms 2,3 is significantly different than the one of Thm 1,2 in [1]. Moreover, from an algorithmic point of view, directly extending [1] to Tucker format would require a computational cost of $O(\sum_i n_i \prod_{j\neq i}r_j)$, which easily becomes prohibitive and poorly performing.
> Could you please clarify why you find the above novelties not significant enough?
>
> 1. Re the lack of computational complexity: We agree and apologize for not adding these to the initial submission. We have provided details about computational complexity and training timing in the global (general) comment above.
>
> 1.  In our experiments we use spectral initialization for low-rank layers, as it is standard in the factorized layer literature. A main strength of our method is that the initialization does not significantly affect convergence, accuracy or the compression rate. This lies in the fact that the careful construction of the method enables us to achieve convergence rates independent of the curvature of the low-rank manifold. Such a result is not achieved by other low-rank approaches which require carefully chosen initializations, though the DLRT method in [1] achieves the same theoretical result, however at prohibitive costs when being applied for tensors. The threshold needs to be chosen by the user and identifies how many singular values need to be kept. This choice depends on the architecture.
>
> 1.   We would like to underline that we do present results for ResNet18/CIFAR10 as well.  Results on ImageNet are currently out of reach, given our limited computational resources. We are currently running tests on TinyImageNet and hope to have the results out soon.
>
> Concerning your question on whether is there any way to further improve efficiency to make the method practical:
>
> We are not sure why you evaluate our method as not practical. As we show in our numerical experiments, TDLRT achieves high compression rates with accuracies comparable to the full-rank model while outperforming current low-rank training methods, including DLRT. There are certainly various approaches to further improve efficiency, including tensor decompositions other than Tucker, using a different framework with more efficient SVD solvers, making use of parallelism in the factor updates. However, the method outperforms current baselines and is already efficient and practical.

---

> > ### Author Response · Authors · 2023-11-16
> > **Details on training behavior**
> >
> > 1. Question: Would be great to provide more details on the training behavior of TDLRT, such as how rank evolves during training, convergence behaviors regarding various compression thresholds, etc.
> >
> >   - Answer: Thank you for the feedback. We provide a showcase of the rank evolutions over the optimization steps of VGG16 on Cifar10 [here](https://imgur.com/gallery/qDyrZM9). We display the ranks of all rank-adaptive tensor bases, i.e., $U_i$ of the whole network, while leaving out fixed rank bases. We can see that in the case of $\tau=0.1`, **within less than half an epoch** (that consists of $391$ optimization steps), **the ranks converge to a steady state**. Similar results can be seen for other values of the compression threshold $\tau=\{0.08, 0.05\}$. For convergence behavior, we kindly refer to Fig. 2 in the main manuscript, where it is explained that TDLRT converges much faster than standard low-rank factorized gradient descent, even for unfavorable initial conditions.
> >
> >
> > 2. Question How often do you perform basis augmentation and tucker decomposition using SVD? If it is performed frequently for every step, does the numerical instability affects the convergence? such that, at the later stage of training, the perturbation on
> >  by learning algorithms may be too small compared to the numerical error induced by SVD.
> >
> >  - Answer: Different strategies to perform the rank truncation are possible. However, in this work an SVD is computed in every update step. We would like to stress that this SVD only needs to be computed for the small core tensor and is independent of the batch size (see the computational costs section in our general comment). Since TDLRT is a stable algorithm, negative effects of rounding errors do not affect the solution in a noticeable way. Note that all operations in network training  such as evaluations, gradient computations and optimizer steps lead to rounding errors. TDLRT is affected by these rounding errors just as any other stable optimization method. Additional rounding errors from the SVD do not amplify general rounding errors in any observable way.

---

### Official Review · Reviewer_T5Ao · 2023-11-13

**Soundness:** 3 good
**Presentation:** 3 good
**Contribution:** 2 fair
**Rating:** 5
**Confidence:** 3

**Summary:**

This paper proposes an adaptive compression methods for convolution layers using the Tucker decomposition. An interesting feature of the paper is the use of Riemannian geometric approach for tensor decomposition allowing to adaptive rank selection.  Experiments are reasonably good. The motivation to use Tucker decomposition over other methods is not clear. Further, the methods employed may not be the optimal decomposition approach.

**Strengths:**

1) The problem is useful and relevant.

2) Motivation experimental results.

**Weaknesses:**

1) The proposed method is incremental work and lack strong novelty.

2) All possible tensor decomposition methods are not considered such as tensor-train. See question 1.

**Questions:**

1) Though Tucker based decomposition of the convolution layers seems to work well, I wonder if tensor-train is better decomposition [1] for this setting. Tensor-train decomposition  avoids the core tensor, while uses fewer ranks compared to Tucker.  Is there a specific advantage of the proposed method over Tucker? Adaptive rank based learning has also been explored with  Riemannian geometry [2].

2) What are the actual training times compared with the proposed model and baseline methods?

[1] A I. V. Oseledets.  Tensor-Train Decomposition. 2011, SIAM Journal on Scientific Computing, 2295-2317

[2] Michael Steinlechner, Riemannian Optimization for High-Dimensional Tensor Completion,  2016,  SIAM Journal on Scientific Computing

---

> ### Author Response · Authors · 2023-11-16
>
> Dear Reviewer,
> thank you for your feedback, comments, and questions.
>
> We agree that the question of which low-rank tensor format to use is a pressing and interesting one.
> As also clearly stated in [2],  the Tucker tensor format is the typical choice for problems involving tensors with $d\leq 5$ modes. Since convolution layers of neural networks are typically order 4, or order 5 tensors, for 2D, respectively 3D convolutions, it is in our view natural to consider Tucker tensors.
> Certainly, it is interesting to consider dynamical low-rank training for higher-order tensor structures using Tensor Trains. However, we remark that it is nontrivial to extend the approach we developed here to Tensor Train format. Thus, we consider it a potential and nontrivial future research direction.
>
> As asked, we would like to point out the major differences with [2]:
>
> - Paper [2] focuses on tensor completion while our algorithm is for neural network training
> - Paper [2] uses Riemannian gradients whereas we propose  a splitting method to integrate the gradient system given by the training problem. Our splitting scheme allows us to integrate each sub-problem (on each factor) efficiently, without needing to expensively compute the whole gradient (please notice also the general comment above); Instead, **computing the Riemannian gradient in general requires to compute the full gradient followed by a projection onto the tangent plane.**  This operation can be very costly and its potential bottleneck for neural network training is not addressed in [2]. Instead, the paper [2] focuses specifically on Tensor completion, which is known to be a particularly favorable setting for computing Riemannian gradients.
>
> - Concerning the rank adaptive strategy proposed in [2], our understanding is that they first run the optimization algorithm until convergence for an initially chosen small rank, and then use the optimizer as a starting point for a new run of the algorithm with an increased rank. This is repeated for several ranks. This approach is used also by other authors, also in the transfer learning community. It certainly is a possibility but has the potential bottleneck of requiring the full training of the network for each of the chosen ranks, which may be computationally unfeasible. We also emphasize that our choice of rank adaptation has the theoretical advantage of approximating the full model, as stated in our Theorem 3 in the submitted paper. This result is not available for the rank adaptive strategy used by [2].

---

### Official Review · Reviewer_AtDm · 2023-11-13

**Soundness:** 2 fair
**Presentation:** 2 fair
**Contribution:** 2 fair
**Rating:** 5
**Confidence:** 2

**Summary:**

To reduce the cost of training convolutional neural networks, this work proposes a low-parametric training method that combines tensor Tucker format and adaptive pruning of Tucker ranks for convolutional layers. Noticeably, the method allows for the adaptivity of the rank of convolutional filters depending on compression rate and robustness for their condition. The authors verify the method's effectiveness by comparing the proposed method with the full training and baselines and observe the cost is reduced due to the proposed method.

**Strengths:**

Reducing the training cost of neural networks has attracted much interest, and many papers proposed the method aiming at practical training with less memory consumption. However, most methods focus on the training cost of a fully connected layer and cannot apply to the convolutional layers. Out of all, the proposed method TDLRT gives the efficiency of training convolutional layers.

**Weaknesses:**

- TDLRT is complicated compared to vanilla SGD. Indeed, the method needs to compute QR decomposition, the projection, etc, causing significant additional costs. Taking this additional computational cost, it is unclear how computationally efficient TDLRT is.

- Theorem 2 does not necessarily indicate the convergence of loss. And there is no comparison with the other method regarding the convergence rate and complexity.

**Questions:**

- It would be nice if the authors could discuss the computational cost per iteration. The computational complexity should also be considered for a fair comparison with SGD and competitors.

- Is it possible to show the convergence of the loss function by training with TDLRT?

---

> ### Author Response · Authors · 2023-11-15
>
> Dear reviewer, thank you for your comments.
>
> **Re the weakness about complications added by TDLRT:**
>
> We appreciate we did not provide enough details about this important point in the paper and we will add them in the new version.
> In particular, we have added details about computational complexity per iteration and training times in the global response as a general comment above.
> TDLRT is a robust low-rank optimization method to train a neural network exploiting manifold information while automatically adapting the ranks of each layer during training. This key property comes a the expense of some methodological and computational complexity. However, **this is in our view a strength not a weakness**.
> One of our main points is exactly that if you perform a "naive" training on the low-rank manifold using a factorized representation of weights and gradients, you struggle to achieve competitive compression/performance ratios, while with the proposed Riemannian scheme one achieves much better performance. The more "complicated" method requires more expensive iterations, but then allows us to drastically reduce the overall cost of training as compared to vanilla factorized SGD, since the method converges much faster (see also Figure 2 in the main manuscript and the new timing tables in the general comment). *Factoring in the additional cost of 5.6 (best case) to 8.3 (worst case) times longer convergence duration of the vanilla approach, the most expensive version of TDLRT is up to 30\% faster. The non-adaptive TDLRT method is up to 4 times faster than the vanilla method when factoring in convergence time.*
>
> **Re the convergence of the method:**
>
> You are correct, Theorem 3 only guarantees that we will be close to a local optimum, if the baseline converges. Deriving a stronger convergence result might require a completely different analysis. We believe that under certain conditions especially on the learning rate such a result can be derived. Indeed, we are aware of ongoing work in this direction for the matrix DLRT case by Arsen Hnatiuk and Andrea Walther. In their analysis, the Robins-Monro conditions on the learning rate need to be assumed. Moreover, the full gradient projected onto the range of the basis at iteration k needs to be bounded by a positive constant plus the projected gradient at the previous time step. We are currently unsure if and how this result extends to the tensor case, but we believe this is a very interesting research direction.
>
> **Re computational complexity:**
>
> We kindly refer to the general response in the global comment above.

---

### Author Response · Authors · 2023-11-15

Dear ACs and Reviewers,

Thank you for your comments and feedback.
As many have shown concerns about novelty and computational efficiency of the proposed method, we present in the next two general comments (a) a comparison of training times for the proposed TDLRT as compared to standard factorized and full  baselines, (b) a computational complexity analysis of each iteration of TDRLT, and (c) a detailed clarification of novelty, in particular with respect to [1]

---

> ### Author Response · Authors · 2023-11-15
> **Training time comparison**
>
> We have implemented a time comparison on ResNet / CIFAR10 which we show in the tables below.
>
>
> **Table 1.**
> We measure the mean time to perform one batch update for various methods. TDLRT is displayed in its rank adaptive and fixed rank versions. Rank adaption comes at the cost of computing an SVD per adaptive tensor mode. While we would like to underline that the SVD is only required for parts of the *small* core tensor, we notice that this is clearly the computational bottleneck.  We remark here, that Pytorch's SVD implementation is not optimally implemented for cuda devices, which is a known issue. Thus we expect improvement for the augmented TDLRT timings with respect to the one presented here with a dedicated cuda implementation of the algorithm. The QR implementation barely affects the computational cost.
>
> | Compression rate [\%] | Full Rank Baseline | Tucker factorization [s] | TDLRT (fixed rank) [s] | TDLRT (rank-adaptive) [s] |
> |-----------|--------------------|-------------|------------------------------------------|------------------------------------|
> | 1         | 0.04               | 0.07        | 0.11                                     | 0.6                                |
> | 0.75      | n.a.               | 0.05        | 0.09                                     | 0.6                                |
> | 0.5       | n.a.               | 0.04        | 0.08                                     | 0.47                               |
> | 0.25      | n.a.               | 0.03        | 0.06                                     | 0.27                               |
> | 0.1       | n.a.               | 0.03        | 0.05                                     | 0.15                               |
> | 0.05      | n.a.               | 0.02        | 0.05                                     | 0.11                               |
>
>
> **Table 2.**
> We measure the time to reach 80% accuracy in the Cifar10 dataset. We see that the faster batch update time of the vanilla method is compensated by a larger time to convergence, thus the TDLRT method is competitive in terms of computational complexity. If the rank adaption is not performed in each batch update, TDLRT is significantly faster than vanilla methods.
>
> | Tucker factorization (min time to conv.) | Tucker factorization (max time to conv.) | TDLRT (rank-adaptive) | TDLRT (fixed rank) |
> |-----------------------------|-----------------------------|---------------------------------|--------------------------------------|
> | 6.552                       | 13.104                      | 28.8                            | 5.44                                 |
> | 5.4432                      | 10.8864                     | 28.8                            | 4.48                                 |
> | 4.032                       | 8.064                       | 22.4                            | 3.84                                 |
> | 3.024                       | 6.048                       | 12.96                           | 2.88                                 |
> | 2.688                       | 5.376                       | 7.2                             | 2.56                                 |
> | 2.352                       | 4.704                       | 5.44                            | 2.4                                  |
>
>
>
> The two tables above show time per iteration and overall training time to reach the best accuracy that the least performing method achieves. While time per iteration favors the direct factorization approach over the proposed TDLRT, due to the additional Riemannian projection step which requires SVD and QR decompositions, as we show also in the paper (Figure 2), direct factorizations converge much slower than the proposed manifold-aware training scheme and thus the overall training time shown in Table 2 above highlights the dramatic advantage gained. We will add this table to the paper, with possibly further numbers on other networks/datasets.
>
> We would like to emphasize that the time per iteration shown is overall expected given the computational complexity analysis that we present in the next comment.

---

> ### Author Response · Authors · 2023-11-15
> **Computational complexity per iteration**
>
> When counting the number of operations, the main costs for the full network come from back and forward passing through the convolutional filters which require $\mathcal{O}\left(b\prod_j n_j\right)$ operations, where $b$ is the batch size. When using TDLRT, these computational costs reduce to $\mathcal{O}\left(b\prod_i r_i + b\sum_i n_i r_i\right)$ operations, yielding a significant reduction in computational costs to determine the gradient. However, the computational costs to perform the low-rank updates also require computing several factorizations. Here, the QR and SVD on $\text{Mat}\_i(C)$ which is needed in the updates of $U_i$ and the truncation step require $\mathcal{O}\left(\sum_i r_i\prod_j r_j\right)$ operations, and the QR on $K_i$ requires $\mathcal{O}(\sum_i n_i r_i^2)$.
>
> Hence, in total, we have for every layer a cost of $C_{DLRT}=\mathcal{O}\left( b\prod_i r_i + b\sum_i n_i r_i + \sum_{i=1}^4 \left(n_i r_i^2 + r_i\prod_{j = 1}^4 r_j\right)  \right)$ operations, vs. the  $C_{Full}=\mathcal{O}(b\prod_i n_i)$ required by the full baseline. Clearly, when $r_i\ll n_i$ we have $C_{DLRT}\ll C_{Full}$.
>
> We would like to underline, that the construction of our method does not require costs of $\mathcal{O}(b \sum_i n_i \prod_{j\neq i}r_j)$ which are the expected costs of the direct extension of DLRT to Tucker tensors, as pointed out in the main paper. Note that our current implementation is not optimal as several steps that can be computed in parallel are performed sequentially. This holds true for updating the basis as well as the truncation of the core.

---

> ### Author Response · Authors · 2023-11-15
> **Novelty**
>
> We notice concerns regarding the novelty of our approach.
> We would like to stress that new techniques and ideas are not only needed for the numerical analysis, but also for the method itself. The DLRT algorithm of [1] can be extended to Tucker tensors in a straightforward manner, yielding an incremental and computationally infeasible training method for convolutional layers. Similar to DLRT, such an extension computes the updated basis $U_i$ by projecting the gradient against all basis functions $U_j$ where $j\neq i$. Then, with the extension of the $K$ used in DLRT to tensors $\mathcal{K}\_i := C \times_i U_i$, we have
> $$
>         \dot{\mathcal{K}}\_i(t) = \nabla \mathcal{L} \bigtimes_{j\neq i} U_j.
> $$
>     This simple extension to Tucker tensors, while clearly preserving the favourable analytic results of the matrix version in [1], is not computationally feasible as evolving $\mathcal{K}\_i$ requires $\mathcal{O}(n_i \prod_{j\neq i}r_j)$ operations per data point. Instead, the idea of this work is to propose a new and non-trivial definition of $K$. Here, as discussed in our work, we choose $K$ based on a carefully chosen QR-decomposition of the core tensor to reduce computational costs in the final resulting algorithm to $\mathcal{O}(n_i r_i)$ per data point (see also the additional detailed observation about gradients below). It is not obvious that this choice of $K$ will actually preserve the favourable properties of DLRT that have been presented analytically and numerically in [1], however we can show that this is the case for the modifications that we propose in this work.
>
> We wish to underline two things:
> - First, this novel methodology is by no means a trivial extension and while we understand that the name TDLRT might make our method appear to be an incremental change to DLRT, a lot of work and ideas were needed to obtain a robust training method for convolutional layers;
> - Second, the extension of DLRT to tensors makes this method available for a large number of modern neural network architectures and is therefore an extremely important contribution for low-rank pruning strategies.
>
> #### **Details about gradients computation**
> To emphasize an additional nontrivial difference with respect to [1], we would like to emphasize here that the computation of the gradients becomes more cumbersome when moving to the tensor setting, and we highlight here how we addressed this issue in a nontrivial way to ensure efficiency of the algorithm (and in particular the cost per iteration discussed in the general comment before). This strategy is hidden in the proof in the appendix but we realize it is worth highlighting it independently and we will do so in the new version of the paper.
>
> The direct use of [1] in the Tucker tensor format would provide a method to compute the network gradients by evaluating them only on the core tensor $C$ and the basis matrices $K_i$. However, while doable in the matrix case, in the case of Tucker tensor format this would require evaluating the network and the gradient tapes $5$ times for an order $4$ tensor, becoming unfeasible.
> We provide the following corollary to significantly reduce the necessary network and gradient tape evaluations. We emphasize that this result also directly applies to and thus improves the matrix case discussed in [1].
>
> ------
>
> **Theorem**.
> Let $W=C\times_{i=1}^4U_i\in \mathcal M_\rho$ and let $\mathrm{Mat}\_i(C)^\top=Q_iS_i^\top$ be the QR decomposition of
> $\mathrm{Mat}\_i(C)^\top$ and let $K_i = U_iS_i$. Then,
> $$
>     \rm{span}\left(\left[U_i,\dot K_i\right]\right) = \rm{span}\left(\left[U_i,\nabla_{U_i}\mathcal L\big(W\big)\right]\right).
> $$
> *Proof:*
> From Eq. (10) in the proof of Theorem 1 in the appendix, it is apparent, that
> $$
>          \dot{K}\_i = \dot{U_i} S_i + U_i \dot{S}\_i = -\mathrm{Mat}\_i(\nabla_W \mathcal{L} \big(\mathrm{Ten}\_i(W)\big))V_i.
> $$
> Moreover, we observe that using the chain rule,
> $$
>         \nabla_{U_i}\mathcal{L} = \mathrm{Mat}\_i(\nabla_W\mathcal{L})\nabla_{U_i}(U_iS_iV_i^{\top})=\mathrm{Mat}\_i(\nabla_W\mathcal{L})V_iS_i^{\top}
> $$
> Combining them, we get
> $$
>         \nabla_{U_i}\mathcal{L}=\mathrm{Mat}\_i(\nabla_W\mathcal{L})V_iS_i^{\top} = -\dot{K}\_i S_i^{\top}.
> $$
>     Using full-rankness of $S_i$ concludes the proof.
>
> ------
>
> Consequently, one can replace the individual forward evaluation and descend steps for $K_i$ in Algorithm 1 by a single network evaluation. All available new information is given by the gradients $\nabla_{U_i}\mathcal L$, which can be evaluated from the same tape. Moreover, we directly obtain that the gradient update given by the theorem above fulfills the local error bound of Theorem 3 in the main paper.
>
>
> [1] Schotthöfer et al. Low-rank lottery tickets: finding efficient low-rank neural networks via matrix differential equations. NeurIPS 2022.

---

> > ### Author Response · Authors · 2023-11-22
> > **Tiny Imagenet**
> >
> > # Results of TDLRT on Tiny-Imagenet with ResNet18
> >
> > Dear reviewers,
> >
> > As promised, we present here the results of TDLRT on Tiny-Imagenet with ResNet18. The ResNet18 architecture is the standard torchvision ResNet18, chosen in accordance with the other ResNet18 models used in this paper. We use SGD with momentum=0.9 for 250 epochs at batch size 128. The learning rate is set to 0.01 with a multi-step scheduler changing the rate after 25 and 40 epochs. The data augmentation consists of random cropping, rotations, horizontal flips, and RGB shifts.
> >
> > ## Test Results
> >
> > | Framework           | Test Accuracy (%) | Compression Rate (%) |
> > |---------------------|-------------------|----------------------|
> > | Baseline            | 51.1              | 0.0                  |
> > | TDLRT $\tau = 0.02$ | 50.90             | 83.95                |
> > | TDLRT $\tau = 0.03$ | 50.73             | 84.40                |
> > | TDLRT $\tau = 0.04$ | 50.66             | 86.82                |
> > | TDLRT $\tau = 0.05$ | 50.73             | 89.94                |
> > | TDLRT $\tau = 0.06$ | 49.32             | 92.12                |
> > | TDLRT $\tau = 0.07$ | 49.04             | 94.05                |
> > | TDLRT $\tau = 0.08$ | 47.58             | 95.31                |
> > | TDLRT $\tau = 0.1$  | 45.43             | 96.85                |
> > | Tucker (60% of full rank)          | 51.06             | 49.20                |
> > | Tucker (50% of full rank)          | 50.42             | 62.51                |
> > | Tucker (40% of full rank)          | 50.17             | 74.18                |
> > | Tucker (30% of full rank)          | 49.66             | 83.66                |
> > | Tucker (20% of full rank)          | 48.39             | 91.08                |
> >
> > The baseline accuracy is 51.1% on average, which is expected for a standard ResNet18 on Tiny-Imagenet. As observed in the other test cases, TDLRT is able to preserve the model's performance at higher compression better than pure gradient descent on decomposed tensors. In particular, at a satisfactory accuracy of ~51%, we observe that TDLRT yields a high compression rate of ~84%, while the compression rate obtained with Tucker is significantly lower at ~50%.
> >
> > Thank you again for your valuable feedback. As the rebuttal deadline nears, we would like to reach out and inquire if our replies have been to your satisfaction. We are happy to clarify any remaining questions that might arise.

---

### Meta-Review · Area_Chair_bhDz · 2023-12-16

**Metareview:**

This paper proposes a new method called TDLRT that execute the Tucker type decomposition based on Riemannian structure to approximate the convolutional filters. The proposed method is combined with rank adaptation procedure. The descent property of the proposed method is theoretically guaranteed. Numerical experiments are executed to show effectiveness of the proposed method.

The proposed method TDLRT is an interesting method with a good empirical performance. On the other hand, the justification of the proposed method could be more theoretically well justified compared with other methods (not only descent property but why it gives a good performance). The numerical experiments can be more extensively executed in larger data sets and with more existing methods such as tensor train method. To address these issues, this paper requires revision before it is accepted. Consequently, I cannot recommend acceptance in the current format.

**Justification For Why Not Higher Score:**

Although the method seems effective based on the numerical experiments, its evaluation can be executed in more thorough way. For example, it can be compared with more state of the art factorization techniques. Moreover, its (theoretical) justification can be extended in a more convincing way.

**Justification For Why Not Lower Score:**

N/A

---

### Decision · Program_Chairs · 2024-01-16

Reject